# Use of Legumes in Extrusion Cooking: A Review

**DOI:** 10.3390/foods9070958

**Published:** 2020-07-20

**Authors:** Antonella Pasqualone, Michela Costantini, Teodora Emilia Coldea, Carmine Summo

**Affiliations:** 1Department of Soil, Plant and Food Science (DISSPA), University of Bari Aldo Moro, Via Amendola, 165/a, I-70126 Bari, Italy; michela.costantini@uniba.it (M.C.); carmine.summo@uniba.it (C.S.); 2Department of Food Engineering, University of Agricultural Sciences and Veterinary Medicine, Calea Manastur, 3-5, 400372 Cluj-Napoca, Romania; teodora.coldea@usamvcluj.ro

**Keywords:** pulses, extrudate, expansion ratio, starch, gelatinization, phytate, α-galactoside, bean, chickpea, pea

## Abstract

The traditional perception that legumes would not be suitable for extrusion cooking is now completely outdated. In recent years, an increasing number of studies have been conducted to assess the behavior of various types of legume flours in extrusion cooking, proving that legumes have excellent potential for the production of extruded ready-to-eat foods by partially or totally replacing cereals. This review identifies the optimal processing conditions for legume-based and legume-added extruded foods, which allow the improvement of the expansion ratio and give the extrudates the spongy and crisp structure expected by consumers. In particular, the effect of the individual processing parameters on the physical-chemical and nutritional properties of the final product is highlighted. The extrusion cooking process, indeed, has a positive effect on nutritional characteristics, because it induces important modifications on starch and proteins, enhancing their digestibility, and reduces the content of trypsin inhibitors, lectins, phytic acid, and tannins, typically present in legumes. Therefore, the extrusion of legume flours is a viable strategy to improve their nutritional features while reducing home preparation time, so as to increase the consumption of these sustainable crops.

## 1. Why Consider Legumes for the Production of Extruded Foods

Consumer demand for ready-to-eat foods is increasing due to the time-saving needs of the modern lifestyle. Extrusion cooking is a technique largely used for the production of several ready-to-eat products, such as crisp expanded snacks (e.g., puffs, runs, collets, etc.), breakfast cereals, instant soups, meat analogues and sport foods [1,2,3]. Extruded foods are able to attract the consumer for their convenience, pleasant appearance and texture [2,4]. The raw materials for extrusion cooking are mostly cereals, due to their good expansion characteristics. However, in addition to providing energy from starch, extruded foods could act as carriers of other nutrients, if enriched with other ingredients [5].

Legumes are a good source of proteins [6], starch, dietary fiber [7], vitamins [8] and minerals [9], and are particularly important when the consumption of animal proteins is restricted due to limited affordability, or religious, dietary and ethical habits. Furthermore, legumes are sustainable crops that are adaptable to marginal lands [10]. In the past, only soybean was used for the development of extruded food products. In recent years, instead, several studies have taken into account the incorporation of other legumes (such as bean, lentil, pea, chickpea, and faba bean) to improve the nutritional value of extruded foods. Nutrient dense extruded multi-legume bars, mixed with whey protein concentrate, honey and palm oil, have been proposed to mitigate malnutrition in developing countries [5,11].

Extrusion cooking technology is also known to reduce the levels of some anti-nutrients contained in legumes [12] such as tannins [13], phytic acid [14], trypsin inhibitors and lectins [15]. In addition, extrusion cooking is able to increase the digestibility of starch and proteins [16].

Extrusion cooking therefore seems to be suitable for producing an array of ready-to-eat legume-added foods. This topic soon attracted the attention of researchers, and their interest increased over time. Indeed, over the past 10 years an increasing number of articles have been published containing the word combinations “extrusion” and “legume”, or “extrusion” and “pulse”, or “extrusion” and the name of a specific legume, as reported in the Scopus scientific database [17] (Figure 1A). Among them, the largest number of studies were conducted on bean and pea, whereas faba bean was the least studied legume (Figure 1B). In addition, as a sign of the growing interest in legumes, the 68th session of the General Assembly of the Organization of the United Nations declared 2016 as the “International Year of Pulses” [18].

Despite several articles reporting the results of single studies regarding the behavior of various types of legume flours in extrusion cooking, there are no reviews available. In this framework, the purpose of this review is to identify the optimal processing conditions for each type of legume, and define the effect of processing parameters on the physical-chemical and nutritional properties of the extruded products.

## 2. Basics of Extrusion Cooking

The development of the single-screw extruder for fast cooking and expanding corn and rice-based snacks dates back to 1946 in the US, followed by twin-screw extruders, introduced in the mid-1970s [1]. High temperature and pressure (up to 200 °C and 20 MPa, respectively) are the usual conditions for extrusion cooking. Raw materials must be properly ground and conditioned at a certain moisture percentage before being fed to the extruder, which is basically composed of one (Figure 2A) or two rotating screws (Figure 2B) fitted in a heated barrel.

In the initial part of the barrel (feeding zone), the raw material is conveyed and mixed by the rotating screw. Then, with the help of shear energy, the material is further kneaded and compressed (kneading zone) and, by friction and additional heating of the barrel, reaches its melting point and plasticizes, particularly in the final part of the machine (cooking zone) [1].

As the plasticized starchy material exits from the die of the extruder, the air bubbles entrapped within the matrix expand due to instant pressure drop. In addition, with the extruded material being heated to temperatures above 100 °C, a moisture flash-off occurs at the exit of the extruder, further improving the puffing effect [16,19]. The expansion ceases upon cooling, when the plasticized matrix becomes glassy and develops a desirable crispy texture. Extrusion cooking, indeed, is an effective means of aerating foods, thereby converting dense, hard materials into lighter and more appealing forms [20]. The quality of the extruded product is therefore defined mostly by its expansion degree [21].

## 3. Optimal Extrusion Cooking Conditions for Legume Flours

The main parameters to be adjusted in the extrusion cooking process are the temperature, the screw speed and the moisture content of the fed ingredients. These parameters strongly influence the characteristics of the extruded product. Therefore, several studies have been carried out to compare sundry combinations of those parameters, in order to point out the optimal conditions for each type of legume (Table 1). These studies focused on the production of: (i) legume-based extruded foods (from 100% legume flour); (ii) legume-added extruded foods, where legume flour is incorporated into a cereal-based extrudate.

Beside the processing parameters, however, also the content of legume flour, its refining degree and particle size, as well as the type of legume, influence the expansion performance of the extrudates. Prior to extrusion, indeed, legumes are milled to flour either using the entire cotyledons or after removal of the hulls (split flours). Hull removal lowers the content of fibers and minerals but improves the expansion during extrusion [22]. Fibers, particularly the insoluble ones [5], surround the air bubbles preventing their maximum expansion, whereas proteins and lipids reduce puffing due to interactions with swelling starch [23]. A direct correlation has been reported between protein and fiber content added to starch-based extruded products and their bulk density [24,25].

In addition, there are differences in the attitude to extrusion of different legumes, based on their compositional features. Compared with lentil and chickpea flours, split yellow pea flour reaches an expansion ratio and bulk density similar to corn meal due to its higher starch and lower protein content [23]. Chickpea extruded snacks, instead, show the lowest expansion properties, which can be attributed to lower starch and higher fat content compared with other pulses [22]. The expansion of extruded products can be increased by using sodium bicarbonate [26], which releases CO_2_ at the conditions used in the extrusion process. It has to be taken into account, however, that the use of sodium bicarbonate causes an undesirable alkalization of the final product.

### 3.1. Optimizing the Processing Conditions for Legume-Based Extruded Foods

Extrusion trials carried out using 100% faba bean flour, varying both the speed of the extruder screw (200 or 300 rpm) and the size of flour particles (<0.5, 1–2, or 2.5 mm), while keeping constant the temperature of die zone at 140 °C, made it possible to point out that the optimal conditions involve high screw speed and a reduced particle size. In particular, the combination of 300 rpm and <0.5 mm resulted in the least hard, most expanded and crispest snack, with sensory properties similar to commercial extruded corn snacks [33], demonstrating that extrusion cooking offers new food applications for legumes, which have not previously shown great economic importance, such as faba bean. The experimental tests involved the use of a twin-screw extruder [33].

A single-screw extruder, instead, was used to study the influence of moisture content, temperature and, again, screw speed, during the production of snacks composed of 100% cowpea flour [32]. Combinations of moisture content ranging from 16 to 24%, die temperature from 160 to 180 °C, and screw speed from 160 to 200 rpm, were compared by Response Surface Methodology (RSM).

The expansion was positively influenced by low moisture content, high die temperature and high screw speed, resulting in less dense and hard extrudates, having higher water absorption index (WAI), water soluble index (WSI) and organoleptic scores [29]. Therefore, the best product was obtained at 16% moisture, 180 °C die temperature and 200 rpm screw speed. Similarly, a relatively low moisture content, high die temperature and high screw speed were found to be the best conditions to extrude desi chickpea with a twin-screw extruder [29]. The effect of moisture content was further confirmed during the production of 100% yellow pea puffed snacks prepared by using a co-rotating twin-screw extruder, operating at 150 °C die temperature and 200 rpm screw speed. Three different feed moisture contents were tested: 14, 16 and 18%. The best results were obtained at the lowest moisture content, resulting in less dense extrudates [20].

In lentil extrudates obtained by varying the die temperature (140, 160, and 180 °C), the screw speed (150, 200 and 250 rpm) and the feed moisture (14, 18 and 22%) [31], the best sensory and nutritional properties, were reached working at 18% moisture, 160 °C die temperature and 200 rpm screw speed. A twin-screw extruder was used.

### 3.2. Optimizing the Processing Conditions for Legume-Added Extruded Foods

Instead of producing 100% legume extrudates, several studies considered the partial replacement of cereals by legume flours, to compensate essential amino acid deficiencies and improve the nutritional features of cereal-based extruded foods. Moreover, these studies focused on pointing out the optimal processing conditions, and, as expected, generally confirmed the findings of extrusion trials where legumes reached 100%.

The effect of feed moisture content (12–16%), barrel temperature (90–110 °C) and screw speed (100–200 rpm) was evaluated during the production of rice-based extrudates containing, among other ingredients, about 9% green pea flour. A twin-screw extruder was used. Optimal products were obtained at high temperature, high screw speed and low moisture content. In particular, 110 °C, 200 rpm and 12% moisture were the best processing conditions, which made it possible to obtain snacks with a high expansion ratio [28].

Similarly, in maize-based extruded snacks containing 30% black bean flour (0.4 mm particle size), the effect of feed moisture content (from 15 to 25%), and screw speed (from 50 to 240 rpm), was evaluated. The best processing conditions, able to ensure high expansion ratio, were 15% moisture and 238 rpm. A single-screw extruder was used [27]. The effect of the same parameters, feed moisture content and screw speed (in the ranges 14–18% and 400–550 rpm, respectively), was studied also during the production of rice-based extruded snacks containing 30% mung bean flour. A twin-screw extruder was used. The best conditions were, again, low feed moisture content (14%), and high screw speed (550 rpm). These conditions ensured the lowest density and hardness, coupled with the highest water absorption index (WAI) and water solubility index (WSI) [34]. Similar findings were observed in trials aimed at obtaining nutritious puffed snacks based on fermented chickpea flour (50%), yogurt, and potato starch, extruded by a twin-screw extruder. Chickpea was fermented to reduce the beany flavor. In these trials, the die temperature was varied in the range of 130–150 °C and screw speed from 266 to 434 rpm. High temperature (140 °C) and elevated screw speed (434 rpm) were found to be optimal for obtaining less dense extrudates [30].

Extrusion cooking also has an effect on color, which is an important feature for the acceptance of food products in general, including the extruded ones. A study carried out specifically to point out the influence of these technologies on color was made by extruding millet-based blends containing 12–28% pigeon pea flour by a single-screw laboratory extruder. Moisture content was in the range of 12–24%, and the extruder operated at different die temperatures (160–200 °C) and screw speeds (100–140 rpm). The evaluation of overall acceptability, which was higher for extrudates having higher lightness (*L**) value, guided the selection of the optimal conditions. The most influential parameter on color, besides the amount of legumes, was the die head temperature. The optimal processing conditions were identified as: 24% moisture content, 171 °C die temperature and 104 rpm screw speed, with 19% legume flour [36]. In addition, during the production of cowpea-based extrudates, redness (*a**) and yellowness (*b**) increased and lightness decreased as die temperature raised [32]. The *a** and *b** indices increase also as moisture content increases, whereas *L** decreases [25]. These findings highlight that the optimal conditions required for a light color, assuming that a darker color cannot be made acceptable by consumers with an adequate communication strategy, are opposite to those needed for ensuring a high expansion ratio, therefore a balance should be reached.

### 3.3. Identification of the Best Level of Addition of Legume Flour

Several studies implemented the extrusion conditions optimized by other authors and focused, instead, on pointing out the best amount of legume flour to be incorporated into cereal-based extruded foods to balance the nutritional features and the sensory and structural properties. Therefore, these studies compared control extruded foods, without legumes, with those prepared at various levels of addition. Amounts of legumes not exceeding 30% were generally considered able to improve the amino acid composition while keeping the sensory properties (hardness, crispiness, lightness) sufficiently similar to controls.

In particular, the addition of germinated chickpea flour to corn-based extrudates was proposed, by using a single-screw extruder operating at 180 °C and 250 rpm screw speed, with 16% moisture [12]. A germination phase was included, as it is able to increase the protein digestibility, essential amino acid availability [37], and linolenic acid content of chickpea flour [38]. Three levels of addition were evaluated: 10, 20, and 30%. It was found that up to 20% germinated chickpea flour could be added without affecting the expansion properties of the corn-based extrudates.

The fortification with either lentil, pea, or chickpea flour was evaluated to improve the nutritional profile of wheat-based extrudates. Levels of addition up to 15% were found to be optimal. A single-screw extruder was used, operating at 180 °C, 210 rpm and 12% moisture [16].

Blends of maize starch with either navy or red bean flours added at levels of 15, 30 or 45% were tested to produce fortified puffed snacks. A twin-screw extruder was used, set at 150 rpm and 160 °C. Levels of up to 30% bean flour resulted in the best extrudates, as crispy as those obtained from maize starch alone but with better nutritional value [4]. In addition, Lima bean flour was tested, by adding at levels of 25, 50 and 75% to corn flour and extruding in a single-screw extruder at 160 °C, 150 rpm and 15.5% moisture. The extrusion of blends was feasible up to a 50% bean inclusion level, which improved the nutritional value of the expanded product. However, at levels of bean addition as low as 25%, a significant decrease in expansion index was observed [23].

After having tested levels of 25, 50 and 75%, the best amount of everlasting pea flour to be added to wheat flour for preparing extruded vegetarian snacks was 50%, able to balance the structural and nutritional properties. A twin-screw extruder was used [35].

In addition to snacks, the production of extruded powders such as baby foods to be reconstituted with water is another common application of extrusion cooking. These foods have less stringent expansion requirements than snacks, therefore can easily incorporate high levels of extruded legume flours. Trials carried out to produce a nutritionally balanced infant food highlighted that the best level of addition of extruded chickpea flour to nixtamalized corn flour was as high as 73%. Such a level ensured a good content of proteins and available lysine, while keeping the sensory features acceptable. The extrusion of chickpea flour was carried out by a single-screw extruder, operating at 150.5 °C and 190.5 rpm [39].

## 4. Effect of Extrusion Cooking Parameters on the Physical-Chemical Properties of the End-Product

The structural characteristics of an extruded food product, such as bulk density (BD) and hardness, are related to the size and number of gas bubbles developed within the expanded rigid starchy matrix. These characteristics, as well as color, are influenced by the level of legume addition and by the parameters of the extrusion process. In Table 2 are summarized the main outcomes of the researches reported in Section 3. In particular, high screw speed and die temperature, coupled to low moisture content, are the best conditions for enhancing expansion degree, while reducing the BD and hardness of the extrudates. Color, instead, is mostly influenced by die temperature. Higher temperatures, indeed, cause Maillard and caramelization reactions, with a consequent increase in browning and redness [36,40].

Hardness is related to the acceptability of the final product, less hard and more expanded extrudates being the most appreciated. In addition, the BD of extruded foods should be as low as possible, indicating a proper increase in volume of the extrudate. Hardness and BD are positively related to each other [28,41,42,43]. Low moisture content reduces both hardness and BD because it raises the friction within the matrix and therefore increases the drag force, resulting in higher temperature and greater pressure on the die [44,45]. High pressure on the die, in turn, causes a greater expansion of the compressed bubbles at the exit of the extruder, and therefore results in a greater puffing of the extrudate [34,46]. Along with low moisture, also high temperature increases the pressure inside the extruder and then positively influences the expansion, reducing both hardness and BD [34,42,47,48,49,50]. An amylose/amylopectin ratio accounting for 1:3–1:4 in the starchy fraction of the raw materials is needed to optimally obtain puffed and crunchy products [51].

Extrusion cooking also has a relevant effect on the pasting properties of starch (Table 3), resulting in gelatinization and dextrinization at an extent depending on the operational parameters adopted. The degree of gelatinization increases at higher temperature and feed moisture, and lower screw speed [52]. A degree of gelatinization of 79% was assessed in extruded blends of cereal and lentil flours processed at 165 °C and 24% moisture [52].

For infant foods and instant soups, a high degree of gelatinization of the starch granules should be achieved. Being pre-gelatinized, these extruded foods are able to form a viscous paste or a thick solution when room temperature, or slightly warm, water is added. In addition, as extruded flours are already gelatinized, they hydrate faster than the raw flour and are highly digestible. Gelatinized starches can also be used as food thickening agents. Therefore, the aim of extrusion cooking is to gelatinize the starch granules, with a limited dextrinization. If an intense shearing action occurs during extrusion, macromolecules may break down to smaller units, affecting the gelatinization process and, consequently, the viscosity. Low shear rate, on the contrary, makes it possible to reach higher viscosity when the extruded flour is rehydrated [53]. Limited levels of lipids (about 2%) act as lubricants, reducing shear and therefore enhancing gelatinization, because they prevent starch from being degraded to dextrins, whereas higher levels of lipids cause a decrease in gelatinization degree due to the formation of amylose–lipid complexes [52].

Pasting properties of starch are measured by using the micro visco-amylograph (MVA). In particular, extruded flours show higher initial viscosity (which is a desirable characteristic for instant flours) than raw flours [53]. Subsequently, raising the temperature in the MVA, peak viscosity is recorded, which should be lower for extruded than not extruded flours, because completely gelatinized materials do not swell anymore [53]. Instead, if the peak viscosity of extruded flour is similar to that of not extruded flour, then starch is still able to swell upon heating (i.e., to gelatinize), indicating that during extrusion only partial gelatinization occurred [53]. However, the presence of constituents other than starch, such as proteins, lipids, and fibers, negatively influences the peak viscosity [54] of legume flours, both raw and extruded, as recently reported in beans [55]. Indeed, in legume flours the starch granules are surrounded by a protein matrix, which limits their hydration and swelling [55]. In addition, high screw speed has a negative effect on gelatinization, lowering the initial viscosity and raising the peak viscosity of extruded flours [55].

Finally, the increase in viscosity at the end of the cooling phase of the MVA analysis is due to the retrogradation of gelatinized starch. Unprocessed flours show a higher degree of retrogradation than extruded ones, which have undergone a relevant thermal and mechanical degradation [53,55]. Therefore, extruded legume flours could be profitably added to bakery products, which could help delaying starch retrogradation.

Another parameter related to starch is the WAI (Table 2), which measures the weight of starch after swelling in excess water [56]. In instant soups and infant foods, WAI should be as high as possible, because it is related to the ability of extruded flours to be easily reconstituted with water in a thick suspension. High WAI is observed at high extrusion temperature [31,53], high moisture content [28], and low screw speed [34,57]. High screw speeds have harsher effects on starch polymers, leading to molecule breakdown and affecting gelatinization and, therefore, ability to bind water [34,57].

WSI is related to the presence of water-soluble molecules deriving from polymer breakdown induced by extrusion. It is used as an indicator of the starch degradation. High WSI results in sticky extrudates [58]. WSI increases with the raise of temperature and screw speed (Table 2), because more severe thermo-mechanical conditions cause a greater extent of dextrinization [43,48,59]. Less severe processing conditions or high content of lipids, which form complexes with amylose, contribute to reduced starch degradation with lower amounts of low molecular weight water-soluble products. Similarly, high feed moisture results in low WSI because moisture plasticizes the extruding material, reducing its viscosity and therefore lowering friction phenomena with a consequent ‘protective’ effect on flour constituents [34].

## 5. Effect of Extrusion Cooking Parameters on the Nutritional Characteristics of the End-Product

The extrusion cooking process has a positive effect on the nutritional characteristics of the end-products, because it induces important modifications on starch and proteins, enhancing their digestibility, and reduces the content of trypsin inhibitors, lectins, phytic acid, and tannins.

### 5.1. Effect on Starch and Proteins

The thermal treatment associated with extrusion cooking is effective in improving protein and starch digestibility compared with traditional thermal processes. Lysine loss may however occur, depending on the extrusion conditions. In particular, starch and protein digestibility take advantage of temperature and feed moisture increases, whereas lysine loss increases as the temperature raises (Table 4).

Humans cannot easily digest non-gelatinized starch [61]. Extrusion cooking results in starch gelatinization, total or partial, at much lower moisture levels (12–22%) than is needed by other processing technologies [61]. The starch digestibility can increase to about 90% by raising the extrusion temperature, which enhances starch gelatinization [31,62]. In addition, extrusion cooking causes a cleavage of amylose and, particularly, of amylopectin molecules induced by the shear, resulting in smaller and more digestible fragments, i.e., dextrins, and reducing sugars [63]. An increase in starch digestibility was observed in lentil-based extrudates by raising extrusion temperature from 140 to 180 °C and feed moisture from 14 to 22% [31]. Starch digestibility can be therefore modulated by regulating the processing parameters of extrusion, because some extruded foods, such as infant flours, have to be highly digestible, whereas others, such as extruded snacks for obese people, should contain little digestible material.

Generally, the digestibility of proteins also increases with extrusion cooking [16,25,31,61]. The denaturation of proteins, in fact, induced by heat and by high friction and shear forces, may improve the accessibility of sites sensitive to proteolysis [64,65]. The surface area also increases, further enhancing the exposure of sites to the enzymes [5]. Compared to the non-extruded raw material, extrusion cooking causes an increase in the in vitro digestibility of proteins by about 13–18% [66], therefore raising the digestibility even above 90%, as was observed in ready-to-eat extrudates based on rice fortified with carob fruit and bean [66]. On the other hand, digestibility can be compromised by the formation of protein aggregates via hydrogen bonds, hydrophobic interactions and disulfide bonds with a consequent decrease in solubility [61,66,67]. In addition, Maillard complexes may be formed during extrusion, particularly at high temperatures and low feed moistures. Maillard reaction particularly affects the bioavailability of lysine, which is limiting in cereals and increases upon legume incorporation, because of the presence of two available amino groups. Furthermore, arginine, tryptophan, cysteine and histidine might also be affected [61]. To lower the incidence of Maillard reaction, mild extrusion conditions should be adopted (<180 °C and >15% moisture) [61].

The improvement of protein digestibility is probably mostly due to the reduction in trypsin inhibitors, which interfere with proteolysis and are heat-labile [25,64,66,68,69], even though in vitro protein digestibility of pea extrudates has been reported to be positively affected by the presence of an increased proportion of globulins vs. albumins [70], also highlighting that the ratio albumin:globulin is influent. In addition, the anti-nutritional factors, such as phytic acid, tannins, and polyphenols, which contribute to lower protein digestibility by linking proteins and decreasing their solubility and susceptibility to proteolysis, are reduced by thermal treatments [71]. Finally, a decrease in insoluble dietary fiber (IDF) in favor of an increase in soluble dietary fiber (SDF) has been observed in extruded bean, pea and lentil-based formulations [66,72,73,74]. Since cell wall rigidity and fiber content may influence the protein digestibility [75], a positive effect of extrusion cooking related to the redistribution of fiber fractions has to be considered.

### 5.2. Effect on Anti-Nutritional Factors and Functional Compounds

Legumes contain several anti-nutritional factors, such as trypsin inhibitors, lectins, tannins, and phytates [64]. Variations in the operative conditions of extrusion cooking influence the content of these compounds (Table 5). An increase in temperature and feed moisture lowers the content of inositol hexaphosphate, trypsin inhibitors and lectins, but has an adverse effect on phenolic compounds and tocopherols. On the other hand, an increase in temperature raises the content of total α-galactosides.

Levels of 15–19 trypsin inhibitor units (TIU) mg^−1^ have been reported in chickpeas, 6–15 TIU mg^−1^ in peas, 5–10 TIU mg^−1^ in faba beans and 3–8 TIU mg^−1^ in lentils [13,76]. The content of trypsin inhibitors markedly decreases after extrusion, due to heat and intense mechanical stress. A reduction accounting for 90% has been observed after the extrusion of lentils [77]. A 95% decrease has been reported by extruding beans at temperatures comprised between 120 and 150 °C, with a total inactivation at 180 °C [60]. Significant destruction of trypsin inhibitors can be achieved by extrusion at elevated temperatures or by increasing residence time when extrusion is done at lower temperatures [78].

Lectins bind sugar branches of the epithelial surface proteins of the digestive tract, resulting in a disruption of the barrier function, which hampers the absorption of nutrients in the gut [79]. Some lectins are toxic, causing vomit and diarrhea [79]. Legume lectins, however, also have anti-viral, anti-fungal, and anti-cancer activity [80]. The content of lectins strongly decreases with extrusion cooking. A 90% decrease was reported after extrusion of lentils [77]. Reductions between 50 and 97% were observed in extruded products containing pea flour [81].

Tannins have the ability to complex and precipitate proteins in aqueous solutions. The condensed tannins of some legumes, such as faba beans, reduce the digestibility of proteins [13,82]. Faba beans, indeed, have the highest tannin content (0.5–24 g kg^−1^) among legumes, followed by beans (0.3–12.6 g kg^−1^), cowpeas (1.4–10.2 g kg^−1^), peas (0.6–10.5 g kg^−1^) and chickpeas (0.6–2.7 g kg^−1^) [83]. The content of tannins of navy beans, chickpeas, cowpeas and lentils can be reduced by extrusion with decreases ranging from 31 to 76% compared to raw legumes [13,84].

Inositol hexaphosphate (IP6), or phytate, is the most abundant inositol phosphate in legume extrudates [85]. IP6 and inositol pentaphosphate (IP5) negatively influence mineral bioavailability, forming complexes with iron, zinc, and calcium. On the other hand, since phytic acid chelates pro-oxidant minerals such as iron, it has an antioxidant effect [86]. In addition, the less phosphorylated forms IP4, IP3, IP2, and IP have a positive role in type 2 diabetes and promote the intestinal absorption of minerals [14]. Extrusion cooking causes a reduction in total phytates, which is essentially due to a decrease in IP6, whereas some of the less phosphorylated forms, particularly IP4 and IP5, show an increase [85]. The decrease in total phytates was mostly imputable to the heat treatment [87], although an effect of feed moisture was also observed. Extrusion of beans at 150°C, with a moisture content of 20%, reduced the total phytate content by about 20–30% [76]. Another study, carried out on navy and red beans, reported a reduction by 44% working at 160 °C and 22% moisture [4]. The content of total phytates significantly decreased during the extrusion of lentils, with greater reductions at 160 than 140 °C [77]. A decrease in total phytates was also observed during the extrusion of faba beans [88].

Phenolic compounds are abundant both in the usual raw materials for extrusion cooking, i.e., cereals, and in legumes. The extrusion conditions influence the overall impact on phenolics: the adoption of low moisture (<14%) and low temperature (<140 °C) can help to retain them [4,81,89,90,91]. However, the effect of extrusion cooking on the various classes of phenolic compounds is controversial: some studies report an increase in anthocyanins [81] and total phenolics [85], whereas other studies report a decrease in anthocyanins [81], a insignificant variation in flavonols [85], and a decrease in total phenolics [4,92,93]. Anthocyanins, in particular, are present in black beans [94] and in black chickpeas [95,96,97]. The decrease in total phenolics was observed in pea/rice extrudates [81] and in starch/navy bean extrudates [4]. In extrudates containing pea, rice and carob flour, instead, an increase in total phenolics was observed [81], highlighting that if phenolics in the starting flour are mostly bound to dietary fiber of the cell walls, as in carob, than the extrusion process, which tends to partially disrupt fiber, can release phenolics [27,98]. In addition, the extrusion cooking prevents the oxidation of phenolic compounds by inactivating the oxidative enzymes responsible for their degradation [27,99]. On the other hand, a detrimental effect of extrusion cooking may be due to the high temperature reached, which affects the phenolic compounds [100]. Therefore, opposite effects take place, whose result mostly depends on the specific characteristics of the raw material.

A marked decrease in total tocopherols content after extrusion was reported [74]. High extrusion temperature negatively affects α-tocopherol, in particular, whereas high moisture lowers the content of γ-tocopherol [101].

Regarding the antioxidant activity of legume-added extruded foods, it has to be highlighted that it is not only due to the presence of antioxidant bioactive compounds contributed by the extruded raw materials, as modified by the extrusion process, but also to antioxidant compounds, such as Maillard reaction products, which may arise from the thermal modifications related to the extrusion cooking process [81]. Therefore, generally the antioxidant activity increases during the extrusion cooking process. In particular, comparing the values of antioxidant activity with those of raw materials, the extrusion process increased the antioxidant activity of green and yellow peas, and chickpea, at an extent varying in the ranges of 27–114, 12–67, and 25–40%, respectively [27].

Other compounds influenced by extrusion cooking are the α-galactosides, such as raffinose, stachyose, and verbascose. These compounds cause flatulence due to the lack of α-galactosidase in the human intestinal mucosa, but at the same time they have a prebiotic effect because are easily fermented by the colonic flora, resulting in the production of short chain fatty acids that stimulate bifidobacterial growth [102,103,104]. It has been observed that extrusion cooking causes a significant increase, up to 85%, in the content of total α-galactosides, compared to the not extruded raw material. The extent of this increase is higher as the extrusion temperature raises. The content of total α-galactosides was higher in lentil extrudates obtained at 160 than 140 °C [77]. These oligosaccharides, indeed, are relatively heat-stable [105]. Therefore, mechanical-structural modifications in the cell walls (such as partial ruptures with increase in porosity) coupled with the increase in surface area, taking place during the extrusion cooking, may probably increase their availability in the extrudates. An increase in the total content of α-galactosides was also reported by Morales et al. [106] during the extrusion of lentil-based formulations. The single α-galactosides, however, may show a different behavior, also according to the legume type. The instant controlled pressure drop, a technique that combines steam pressure and heat, similar to extrusion cooking, caused an increase in stachyose in lentil, opposed to a decrease in chickpea; further, a decrease in raffinose was observed both in lentil and chickpea, and an increase in verbascose (which was absent in chickpea) in lentil [105].

From the technical point of view, extruding cereal–legume formulations could be therefore a good strategy to produce shelf-stable ready-to-eat nutritious food products. This is of particular importance in geographic areas where is needed to relief malnutrition. However, the cost of purchasing and operating the extruders may be not affordable in developing countries [78]. A solution to this situation may be the application of simple, autogenic single-screw extruders, which are still available on the markets.

## 6. Conclusions

Legumes have shown excellent potential for the production of extruded ready-to-eat foods by partially or totally replacing cereals. The traditional perception that legumes would not be suitable for extrusion cooking is now completely outdated. By accurately selecting the optimal processing parameters, it is possible to improve the expansion ratio and give the extrudates the spongy and crisp structure expected by consumers. Moreover, the addition of legume flours improves the nutritional value of cereal-based end-products by increasing the content of essential aminoacids, fibers, proteins, and micronutrients, while extrusion cooking inactivates the nutritionally undesirable compounds typically present in legumes.

Therefore, the extrusion of legumes is a viable strategy to add value to underexploited legumes and reduce home preparation time, so as to increase the consumption of these sustainable crops.

## Figures and Tables

**Figure 1 foods-09-00958-f001:**
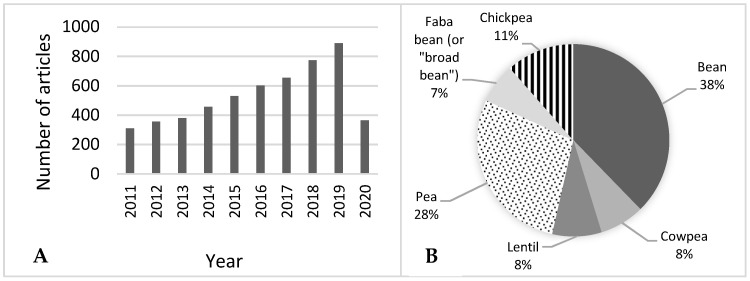
Number of articles regarding the extrusion cooking of legumes published from January 2011 to April 2020 (**A**) and percent distribution of articles according to legume type (**B**). (Elaboration of data from the Scopus database).

**Figure 2 foods-09-00958-f002:**
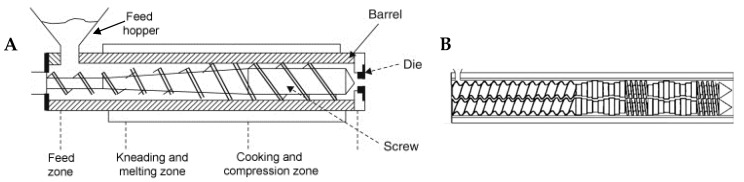
Schematic representation of a cooking extruder with a single screw (**A**) and detail of twin screws (**B**).

**Table 1 foods-09-00958-t001:** The main results of the studies aimed at identifying the optimal processing conditions for the extrusion cooking of legumes. The optimal values for each parameter, among the tested ones, are those highlighted in bold.

Legume Type	Extruder Type	Tested Values					Quality Indices Considered for Optimization	Reference
		Legume Content (g/100 g)	Flour Particle Size (mm)	Maximum Temperature (Die Zone) (°C)	Feed Moisture (g/100 g)	Screw Speed (rpm)		
Bean (*Phaseolus vulgaris* L.)	SS	30	0.4	120–170, **157** ^1^	**15**–25	50–240, **238**	ER, BD, H	[27]
	TS	15, **30**, 45	0.5	160	22	150	ER, H, NP, C	[4]
	SS	**25**, 50, 75	0.5	160	15.5	150	ER, BD, H, NP	[23]
Pea (*Pisum sativum* L.)	TS	100	NS	150	**14**, 16, 18	200	ER, BD	[20]
	TS	9	NS	90, 100, **110** ^1^	**12**, 14, 16	100, 150, **200**	ER, BD, H, WAI, WSI	[28]
	SS	5, 10, **15**	NS	180	12	210	NP	[16]
Chickpea (*Cicer arietinum* L.)		100	NS	110, 120, 135, **150**	19, 20, **22**, 24, 26	260, **300**, 340	ER, BD	[29]
	TS	50	1.25	130, **140**, 150	17	266, 300, 350, 400, **434**	ER, H, SP, NP	[30]
	SS	10, **20**, 30	0.25	160	16	250	ER, BD, WAI, WSI, NP	[12]
	SS	5, 10, **15**	NS	180	12	210	NP	[16]
Lentil (*Lens culinaris* L.)	TS	100	0.25	140, **160**, 180	14, **18**, 22	150, **200**, 250	ER, BD, H, WAI, WSI, SP, NP	[31]
	SS	5, 10, **15**	NS	180	12	210	NP	[16]
Cowpea (*Vigna unguiculata* L.)	SS	100	0.8	160– **180**	**16**–24	160– **200**	BD, H, WAI, WSI, SP	[32]
Faba bean (*Vicia faba* L.)	TS	100	**0.5**, 1.5, 2.5	140	NS	200, **300**	ER, H, SP	[33]
Mung bean (*Vigna radiate* L.)	TS	30	0.2	130–170, **148** ^1^	**14**–18	400– **550**	BD, H, WAI, WSI	[34]
Everlasting pea (*Lathyrus sativus*)	TS	35, **50**, 65	0.6	110, 140, 180, 170, 130	**18**, 21, 24	75	ER, BD, NP	[35]
Pigeon pea (*Cajanus cajan* L.)	SS	19	0.07	160–200, **171**	12– **24**	100–140, **104**	C	[36]

^1^ barrel temperature. SS = single-screw; TS = twin-screw; NS = not specified; ER = expansion ratio; BD = bulk density; H = hardness; WAI = water absorption index; WSI = water solubility index; SP = sensory properties; NP = nutritional properties; C = color.

**Table 2 foods-09-00958-t002:** Effect (positive/negative) of the increase in the main processing parameters of extrusion cooking on the physical-chemical characteristics of legume extrudates.

	Expansion Ratio	Bulk Density	Hardness	WAI	WSI	Color		
						Lightness (*L**)	Redness (*a**)	Yellowness (*b**)
Legume content	−	+	+	NS	NS	−	+	+
Temperature	+	−	−	+	+	−	+	+
Feed moisture	−	+	+	+	−	−	+	−
Screw speed	+	−	−	−	+	NS	NS	NS

WAI = water absorption index; WSI = water solubility index; NS = not studied.

**Table 3 foods-09-00958-t003:** Effect (positive/negative) of the increase in the main processing parameters of extrusion cooking on the starch characteristics of legume extrudates.

	Starch Pasting Properties			Degree of Gelatinization
	Initial Viscosity	Peak Viscosity	Final Viscosity	
Legume content	+	−	−	NS
Temperature	+	−	−	+
Feed moisture	−	−	−	+
Screw speed	−	+	+	−

NS = not studied.

**Table 4 foods-09-00958-t004:** Effect (positive/negative) of the increase in the main processing parameters of extrusion cooking on the nutritional characteristics of legume-based extruded products.

	Protein Digestibility	Starch Digestibility	Lysine Loss
Temperature	+ ^1^	+ ^2^	+ ^3^
Feed moisture	+ ^1^	+ ^2^	− ^3^
Screw speed	+	+	NS

^1^ Temperature = 150–160 °C and moisture = 20–22% are optimal for lentil [25,31], horsegram [25] and pinto bean [60]; ^2^ Temperature = 180 °C and moisture = 22% are optimal for lentil [31]; ^3^ Temperature <180 °C and moisture >15% are generally optimal for all raw materials [61]. NS = not studied.

**Table 5 foods-09-00958-t005:** Effect (positive/negative) of the increase in the main processing parameters of extrusion cooking on the content of functional and anti-nutritional compounds of legume-based extruded products.

Parameter	Phenolics	Tocopherols	Antioxidant Activity	α-Galactosides	Anti-Nutritional Compounds			
					Trypsin Inhibitors	Phytate (IP6)	Tannins	Lectins
Temperature	−	−	+	+	−	−	NS	−
Feed moisture	−	−	NS	NS	−	−	−	−
Screw speed	NS	NS	NS	NS	−	−	−	NS

NS = not studied; IP6 = inositol hexaphosphate.

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
