# Peer review of "Use of Legumes in Extrusion Cooking: A Review"

_foods, 2020, doi:10.3390/foods9070958_

Round 1
Reviewer 1 Report
Although modern trends are directed towards the consumption of moderately processed foods, the trends of the food industry are completely different. The authors of the reviewed publication indicated in the "References" section well over 50% of current publications from 2010–2010. According to my subjective opinion, interesting comprehensive review by Dr. Antonella Pasqualone and colleagues, entitled “Use of legumes in extrusion cooking: a review” of this form neds minor corrections, which I have listed below:
1) At line 37 is: … with low levels of fats, mostly poly- and monounsaturated … . The sentence is unclear. What contains low levels of fats, especially those needed unsaturated? Maybe it's better to delete this sequence.
2) At lines 53 and 54 is: … (Fig. 1) … and (Fig. 2) … . Unfortunately, the review lacks Fig. 2.
3) At lines 63–65 the description is complicated. Please correct using concise wording. Example, such: … published from 01-2011 to 04-2020 (left, A) … (right, B) … . Then the last sentence in brackets is to be deleted and in lines 53 and 54 can be corrected as (Fig. 1A) and (Fig. 1B), respectively, after entering the letters A and B in figure 1.
4) In lines 66–76, in paragraph 2, the authors provided an extremely detailed technical description of the operation of extrusion cooking equipment, but readers would like to see a simplified diagram of these machines.
5) At line 84 is slang: … the molecular changes … . Please explain or alternatively remove sentence … the molecular changes and …, because changes in the structure of chemical molecules are the result of chemical reactions mentioned nearby.
6) In line 108 at the end of paragraph 3, the authors' cautions is required regarding undesirable alkalization of the final product as a result of remaining in the product alkaline sodium species, an unwanted by-product of thermal decomposition of sodium bicarbonate, because, for people with certain gastric dysfunction of highly processed food products, especially alkalized ones, are not recommended. Reference to source literature is also necessary here.
7) In table 1 in the last column (on the right) all references should be in square brackets ( [ ] ). Please corrected, because with such a large amount of data, studying this table is seriously difficult.
8) At Table 1, lines 136–137, authors mistakenly used a short pause sign ( - ) between numbers. The mean pause sign ( – ) should be used, as it is in the lines 150 (twice), 160, 161, 176 (twice), etc. Similar errors are in the lines: 31, 64, 111, 169, 174, 175, 241, 246, 247, 319, 351, 381, 382 (twice at the end of the line), 402, 406, 431, 476, 479, 481, 483, 485, 488, 491, 404, 496, 498, 502, 507, 510, 521, 526, 529, 531, 534, 537, 540, 543, 545, 547, 550, 553, 557, 562, 564, 566, 569, 572, 575, 577, 580, 583, 589, 592, 603, 605, 607, 609, 612, 616, 624, 626, 628, 630, 634, 637, 652, 655, 659, 665, 673, 679, 681, 685, 688, 696, 701, 707, 711, 720, 724, 727, 742, 748, 750, 753, 759, 763 and 767. Please correct - standardize the type of characters used throughout the review.
9) At line 317 is: … Metalchem Gliwice, … , but should be: … Metalchem, Gliwice, … . Please enter a forgotten comma ( , ).
Author Response
Reviewer 1
Although modern trends are directed towards the consumption of moderately processed foods, the trends of the food industry are completely different. The authors of the reviewed publication indicated in the "References" section well over 50% of current publications from 2010–2010. According to my subjective opinion, interesting comprehensive review by Dr. Antonella Pasqualone and colleagues, entitled “Use of legumes in extrusion cooking: a review” of this form needs minor corrections, which I have listed below:
Response: We thank very much the Reviewer for appreciating our work.
1) At line 37 is: … with low levels of fats, mostly poly- and monounsaturated … . The sentence is unclear. What contains low levels of fats, especially those needed unsaturated? Maybe it's better to delete this sequence.
Response: Sorry for lack of clarity. We agree, the sentence has been deleted.
2) At lines 53 and 54 is: … (Fig. 1) … and (Fig. 2) … . Unfortunately, the review lacks Fig. 2.
Response: We apologize for this mistake. We amended the numbering of figures. Fig. 1 is actually composed of two subfigures: 1A and 1B. Figure 2 was actually Fig. 1B.
3) At lines 63–65 the description is complicated. Please correct using concise wording. Example, such: … published from 01-2011 to 04-2020 (left, A) … (right, B) … . Then the last sentence in brackets is to be deleted and in lines 53 and 54 can be corrected as (Fig. 1A) and (Fig. 1B), respectively, after entering the letters A and B in figure 1.
Response: Thanks for suggestion. We amended as suggested.
4) In lines 66–76, in paragraph 2, the authors provided an extremely detailed technical description of the operation of extrusion cooking equipment, but readers would like to see a simplified diagram of these machines.
Response: A simplified diagram of the extruders has been added (Fig. 2A and 2B).
5) At line 84 is slang: … the molecular changes … . Please explain or alternatively remove sentence … the molecular changes and …, because changes in the structure of chemical molecules are the result of chemical reactions mentioned nearby.
Response: The sentence has been removed.
6) In line 108 at the end of paragraph 3, the authors' cautions is required regarding undesirable alkalization of the final product as a result of remaining in the product alkaline sodium species, an unwanted by-product of thermal decomposition of sodium bicarbonate, because, for people with certain gastric dysfunction of highly processed food products, especially alkalized ones, are not recommended. Reference to source literature is also necessary here.
Response: A reference has been added. Moreover, a sentence to highlight that undesirable alkalization of the final product would occur has been added (lines 111-112).
7) In table 1 in the last column (on the right) all references should be in square brackets ( [ ] ). Please corrected, because with such a large amount of data, studying this table is seriously difficult.
Response: Thanks for suggestion. We amended as suggested.
8) At Table 1, lines 136–137, authors mistakenly used a short pause sign ( - ) between numbers. The mean pause sign ( – ) should be used, as it is in the lines 150 (twice), 160, 161, 176 (twice), etc. Similar errors are in the lines: 31, 64, 111, 169, 174, 175, 241, 246, 247, 319, 351, 381, 382 (twice at the end of the line), 402, 406, 431, 476, 479, 481, 483, 485, 488, 491, 404, 496, 498, 502, 507, 510, 521, 526, 529, 531, 534, 537, 540, 543, 545, 547, 550, 553, 557, 562, 564, 566, 569, 572, 575, 577, 580, 583, 589, 592, 603, 605, 607, 609, 612, 616, 624, 626, 628, 630, 634, 637, 652, 655, 659, 665, 673, 679, 681, 685, 688, 696, 701, 707, 711, 720, 724, 727, 742, 748, 750, 753, 759, 763 and 767. Please correct - standardize the type of characters used throughout the review.
Response: Sorry for mistake. We amended all the listed wrong pause signs and standardized the type of characters.
9) At line 317 is: … Metalchem Gliwice, … , but should be: … Metalchem, Gliwice, … . Please enter a forgotten comma ( , ).
Response: This detail, which was related to the Company name of the specific extruder used, has been deleted, as requested by Reviewer no. 3.
Reviewer 2 Report
The work is well prepared, based on current and satisfactorily selected literature sources. The content practically raises the most important issues, which may help in the further and worth recommending dissemination of production of legume-enriched extrudates.
Two minor remarks:
- Table 4 can be further supplemented with more specific data, not just + or -;
- the statement contained in verses 445 - 450 can be supplemented with the suggestion that a certain solution to this situation may be the application of simple, autogenic single-screw extruders, which are still available on the markets.
Suggestion for the future: Authors should think about a similar literature review regarding the processing of fibrous raw materials by extrusion-cooking.
Author Response
The work is well prepared, based on current and satisfactorily selected literature sources. The content practically raises the most important issues, which may help in the further and worth recommending dissemination of production of legume-enriched extrudates.
Response: We thank very much the Reviewer for appreciating our work.
Two minor remarks:
- Table 4 can be further supplemented with more specific data, not just + or -;
Response: Thanks for suggestion. Table 4 has been supplemented with more specific data.
- the statement contained in verses 445 - 450 can be supplemented with the suggestion that a certain solution to this situation may be the application of simple, autogenic single-screw extruders, which are still available on the markets.
Response: The suggested solution has been mentioned (see lines 447-448).
Suggestion for the future: Authors should think about a similar literature review regarding the processing of fibrous raw materials by extrusion-cooking.
Response: Thanks for this suggestion, we will do.
Reviewer 3 Report
Overall, this is quite a nice manuscript. It addresses many relevant topics and sub-topics to the overall topic of extrusion technology.
There are only a couple of very minor points that I would like the authors to look at again in a resubmission.
In the ABSTRACT (line 12) replace the words old idea with traditional perception
Add a reference at the end of the sentence in line 35.
Add a reference with regards to 'adaptability to marginal lands' (line 39)
Please fix figure 1. Refer to the two sub-figures as 1a and 1b, not 'left' and 'right'. Also, the graph on the 'right' is referred to in the text as figure 2 - please fix.
The rest is quite okay from a language point of view. I might have written some sentences differently, but that is purely a personal touch.
I would encourage the authors to remove all mentioning of specific brands when it comes to the single or twin screw extruders. The brands are irrelevant for a scientific paper. I get it that the authors want to point out that there are differences between single screw and twin screw systems. Keep doing that, but remove the references to specific brands (some people might see their mentioning as a commercial endorsement).
Author Response
Overall, this is quite a nice manuscript. It addresses many relevant topics and sub-topics to the overall topic of extrusion technology.
We thank the Reviewer for appreciating our work.
There are only a couple of very minor points that I would like the authors to look at again in a resubmission.
In the ABSTRACT (line 12) replace the words old idea with traditional perception.
Response. Thanks for suggestion. We replaced “old idea” with “traditional perception”.
Add a reference at the end of the sentence in line 35.
Response. A reference has been added at the end of the sentence in line 35.
Add a reference with regards to 'adaptability to marginal lands' (line 39)
Response. A reference with regards to 'adaptability to marginal lands' has been added. Please note that the total count of references in the final list has not increased because we deleted 2 references (one included in a sentence deleted to fulfill Reviewer 1 suggestions, and another deleted because referred to a manuscript from our research group submitted at the time of writing this article but still without response, neither positive nor negative, from the Editor).
Please fix figure 1. Refer to the two sub-figures as 1a and 1b, not 'left' and 'right'. Also, the graph on the 'right' is referred to in the text as figure 2 - please fix.
Response. Figure 1 has been fixed by adding “A” and “B” in the two sub-figures and by referring to 1A and 1B.
The rest is quite okay from a language point of view. I might have written some sentences differently, but that is purely a personal touch.
I would encourage the authors to remove all mentioning of specific brands when it comes to the single or twin screw extruders. The brands are irrelevant for a scientific paper. I get it that the authors want to point out that there are differences between single screw and twin screw systems. Keep doing that, but remove the references to specific brands (some people might see their mentioning as a commercial endorsement).
Response: Thanks for your kind advise. We followed it. At this regard, actually at the time of writing we were uncertain if give or not the details about the specific brands. Finally we opted to give them to ensure a full reproducibility of results, but it is true that it can seem a commercial endorsement. Therefore we have now deleted all of them, because if a reader is particularly interested in knowing also the brand of the extruders can simply read the original reference, which is always given in square brackets in each sentence.